# Exponential Concentration of a Density Functional Estimator

**Shashank Singh**
Statistics & Machine Learning Departments
Carnegie Mellon University
Pittsburgh, PA 15213
sss1@andrew.cmu.edu

**Barnabás Póczos**
Machine Learning Department
Carnegie Mellon University
Pittsburgh, PA 15213
bapoczos@cs.cmu.edu

## Abstract

We analyze a plug-in estimator for a large class of integral functionals of one or more continuous probability densities. This class includes important families of entropy, divergence, mutual information, and their conditional versions. For densities on the $d$-dimensional unit cube $[0,1]^d$ that lie in a $\beta$-Hölder smoothness class, we prove our estimator converges at the rate $O\left(n^{-\frac{\beta}{\beta+d}}\right)$. Furthermore, we prove the estimator is exponentially concentrated about its mean, whereas most previous related results have proven only expected error bounds on estimators.

## 1 Introduction

Many important quantities in machine learning and statistics can be viewed as integral functionals of one of more continuous probability densities; that is, quanitities of the form

$$F(p_1, \cdots, p_k) = \int_{\mathcal{X}_1 \times \cdots \times X_k} f(p_1(x_1), \ldots, p_k(x_k)) \, d(x_1, \ldots, x_k),$$

where $p_1, \cdots, p_k$ are probability densities of random variables taking values in $\mathcal{X}_1, \cdots, X_k$, respectively, and $f : \mathbb{R}^k \to \mathbb{R}$ is some measurable function. For simplicity, we refer to such integral functionals of densities as 'density functionals'. In this paper, we study the problem of estimating density functionals. In our framework, we assume that the underlying distributions are not given explicitly. Only samples of $n$ independent and identically distributed (i.i.d.) points from each of the unknown, continuous, nonparametric distributions $p_1, \cdots, p_k$ are given.

### 1.1 Motivations and Goals

One density functional of interest is Conditional Mutual Information (CMI), a measure of conditional dependence of random variables, which comes in several varieties including Rényi-$\alpha$ and Tsallis-$\alpha$ CMI (of which Shannon CMI is the $\alpha \to 1$ limit case). Estimating conditional dependence in a consistent manner is a crucial problem in machine learning and statistics; for many applications, it is important to determine how the relationship between two variables changes when we observe additional variables. For example, upon observing a third variable, two correlated variables may become independent, and, similarly, two independent variables may become dependent. Hence, CMI estimators can be used in many scientific areas to detect confounding variables and avoid infering causation from apparent correlation [19, 16]. Conditional dependencies are also central to Bayesian network learning [7, 34], where CMI estimation can be used to verify compatibility of a particular Bayes net with observed data under a local Markov assumption.

Other important density functionals are divergences between probability distributions, including Rényi-$\alpha$ [24] and Tsallis-$\alpha$ [31] divergences (of which Kullback-Leibler (KL) divergence [9] is the

$\alpha \to 1$ limit case), and $L_p$ divergence. Divergence estimators can be used to extend machine learning algorithms for regression, classification, and clustering from the standard setting where inputs are finite-dimensional feature vectors to settings where inputs are sets or distributions [22, 18]. Entropy and mutual information (MI) can be estimated as special cases of divergences. Entropy estimators are used in goodness-of-fit testing [5], parameter estimation in semi-parametric models [33], and texture classification [6], and MI estimators are used in feature selection [20], clustering [1], optimal experimental design [13], and boosting and facial expression recognition [25]. Both entropy and mutual information estimators are used in independent component and subspace analysis [10, 29] and image registration [6]. Further applications of divergence estimation are in [11].

Despite the practical utility of density functional estimators, little is known about their statistical performance, especially for functionals of more than one density. In particular, few density functional estimators have known convergence rates, and, to the best of our knowledge, no finite sample exponential concentration bounds have been derived for general density functional estimators. One consequence of this exponential bound is that, using a union bound, we can guarantee accuracy of multiple estimates simultaneously. For example, [14] shows how this can be applied to optimally analyze forest density estimation algorithms. Because the CMI of variables $X$ and $Y$ given a third variable $Z$ is zero if and only $X$ and $Y$ are conditionally independent given $Z$, by estimating CMI with a confidence interval, we can test for conditional independence with bounded type I error probabilty.

**Our main contribution** is to derive convergence rates and an exponential concentration inequality for a particular, consistent, nonparametric estimator for large class of density functionals, including conditional density functionals. We also apply our concentration inequality to the important case of Rényi-$\alpha$ CMI.

## 1.2 Related Work

Although lower bounds are not known for estimation of general density functionals (of arbitrarily many densities), [2] lower bounded the convergence rate for estimators of functionals of a single density (e.g., entropy functionals) by $O\left(n^{-4\beta/(4\beta+d)}\right)$. [8] extended this lower bound to the two-density cases of $L_2$, Rényi-$\alpha$, and Tsallis-$\alpha$ divergences and gave plug-in estimators which achieve this rate. These estimators enjoy the parametric rate of $O\left(n^{-1/2}\right)$ when $\beta > d/4$, and work by optimally estimating the density and then applying a correction to the plug-in estimate. In contrast, our estimator undersmooths the density, and converges at a slower rate of $O\left(n^{-\beta/(\beta+d)}\right)$ when $\beta < d$ (and the parametric rate $O\left(n^{-1/2}\right)$ when $\beta \geq d$), but obeys an exponential concentration inequality, which is not known for the estimators of [8].

Another exception for $f$-divergences is provided by [17], using empirical risk minimization. This approach involves solving an $\infty$-dimensional convex minimization problem which be reduced to an $n$-dimensional problem for certain function classes defined by reproducing kernel Hilbert spaces ($n$ is the sample size). When $n$ is large, these optimization problems can still be very demanding. They studied the estimator's convergence rate, but did not derive concentration bounds.

A number of papers have studied $k$-nearest-neighbors estimators, primarily for Rényi$\alpha$ density functionals including entropy [12], divergence [32] and conditional divergence and MI [21]. These estimators work directly, without the intermediate density estimation step, and generally have proofs of consistency, but their convergence rates and dependence on $k$, $\alpha$, and the dimension are unknown. One exception for the entropy case is a $k$-nearest-neighbors based estimator that converges at the parametric rate when $\beta > d$, using an ensemble of weak estimators [27].

Although the literature on dependence measures is huge, few estimators have been generalized to the conditional case [4, 23]. There is some work on testing conditional dependence [28, 3], but, unlike CMI estimation, these tests are intended to simply accept or reject the hypothesis that variables are conditionally independent, rather than to measure conditional dependence. Our exponential concentration inequality also suggests a new test for conditional independence.

This paper continues a line of work begin in [14] and continued in [26]. [14] proved an exponential concentration inequality for an estimator of Shannon entropy and MI in the 2-dimensional case. [26] used similar techniques to derive an exponential concentration inequality for an estimator of Rényi-$\alpha$ divergence in $d$ dimensions, for a larger family of densities. Both used plug-in estimators

based on a mirrored kernel density estimator (KDE) on $[0, 1]^d$. Our work generalizes these results to a much larger class of density functionals, as well as to conditional density functionals (see Section 6). In particular, we use a plug-in estimator for general density functionals based on the same mirrored KDE, and also use some lemmas regarding this KDE proven in [26]. By considering the more general density functional case, we are also able to significantly simplify the proofs of the convergence rate and exponential concentration inequality.

**Organization**

In Section 2, we establish the theoretical context of our work, including notation, the precise problem statement, and our estimator. In Section 3, we outline our main theoretical results and state some consequences. Sections 4 and 5 give precise statements and proofs of the results in Section 3. Finally, in Section 6, we extend our results to conditional density functionals, and state the consequences in the particular case of Rényi-$\alpha$ CMI.

## 2 Density Functional Estimator

### 2.1 Notation

For an integer $k$, $[k] = \{1, \cdots, k\}$ denotes the set of positive integers at most $k$. Using the notation of multi-indices common in multivariable calculus, $\mathbb{N}^d$ denotes the set of $d$-tuples of non-negative integers, which we denote with a vector symbol $\vec{\cdot}$, and, for $\vec{i} \in \mathbb{N}^d$,

$$D^{\vec{i}} := \frac{\partial^{|\vec{i}|}}{\partial^{i_1} x_1 \cdots \partial^{i_d} x_d} \quad \text{and} \quad |\vec{i}| = \sum_{k=1}^{d} i_k.$$

For fixed $\beta, L > 0$, $r \geq 1$, and a positive integer $d$, we will work with densities in the following bounded subset of a $\beta$-Hölder space:

$$C_{L,r}^{\beta}([0,1]^d) := \left\{ p : [0,1]^d \to \mathbb{R} \left| \sup_{\substack{x \neq y \in D \\ |\vec{i}| = \ell}} \frac{|D^{\vec{i}} p(x) - D^{\vec{i}} p(y)|}{\|x - y\|^{(\beta - \ell)}} \right. \right\}, \tag{1}$$

where $\ell = \lfloor \beta \rfloor$ is the greatest integer *strictly* less than $\beta$, and $\| \cdot \|_r : \mathbb{R}^d \to \mathbb{R}$ is the usual $r$-norm. To correct for boundary bias, we will require the densities to be nearly constant near the boundary of $[0, 1]^d$, in that their derivatives vanish at the boundary. Hence, we work with densities in

$$\Sigma(\beta, L, r, d) := \left\{ p \in C_{L,r}^{\beta}([0,1]^d) \left| \max_{1 \leq |\vec{i}| \leq \ell} |D^{\vec{i}} p(x)| \to 0 \text{ as } \text{dist}(x, \partial[0,1]^d) \to 0 \right. \right\}, \tag{2}$$

where $\partial[0,1]^d = \{x \in [0,1]^d : x_j \in \{0,1\} \text{ for some } j \in [d]\}$.

### 2.2 Problem Statement

For each $i \in [k]$ let $X_i$ be a $d_i$-dimensional random vector taking values in $\mathcal{X}_i := [0,1]^{d_i}$, distributed according to a density $p_i : \mathcal{X} \to \mathbb{R}$. For an appropriately smooth function $f : \mathbb{R}^k \to \mathbb{R}$, we are interested in a using random sample of $n$ i.i.d. points from the distribution of each $X_i$ to estimate

$$F(p_1, \ldots, p_k) := \int_{\mathcal{X}_1 \times \cdots \times \mathcal{X}_k} f(p_1(x_1), \ldots, p_k(x_k)) \, d(x_1, \ldots, x_k). \tag{3}$$

### 2.3 Estimator

For a fixed bandwidth $h$, we first use the mirrored kernel density estimator (KDE) $\hat{p}_i$ described in [26] to estimate each density $p_i$. We then use a plug-in estimate of $F(p_1, \ldots, p_k)$.

$$F(\hat{p}_1, \ldots, \hat{p}_k) := \int_{\mathcal{X}_1 \times \cdots \times \mathcal{X}_k} f(\hat{p}_1(x_1), \ldots, \hat{p}_k(x_k)) \, d(x_1, \ldots, x_k).$$

Our main results generalize those of [26] to a broader class of density functionals.

## 3  Main Results

In this section, we outline our main theoretical results, proven in Sections 4 and 5, and also discuss some important corollaries.

We decompose the estimatator's error into bias and a variance-like terms via the triangle inequality:

$$|F(\hat{p}_1,\ldots,\hat{p}_k) - F(p_1,\ldots,p_k)| \leq \underbrace{|F(\hat{p}_1,\ldots,\hat{p}_k) - \mathbb{E}F(\hat{p}_1,\ldots,\hat{p}_k)|}_{\text{variance-like term}}$$
$$+ \underbrace{|\mathbb{E}F(\hat{p}_1,\ldots,\hat{p}_k) - F(p_1,\ldots,p_k)|}_{\text{bias term}}.$$

We will prove the "variance" bound

$$\mathbb{P}\left(|F(\hat{p}_1,\ldots,\hat{p}_k) - \mathbb{E}F(\hat{p}_1,\ldots,\hat{p}_k)| > \varepsilon\right) \leq 2\exp\left(-\frac{2\varepsilon^2 n}{C_V^2}\right) \tag{4}$$

for all $\varepsilon > 0$ and the bias bound

$$|\mathbb{E}F(\hat{p}_1,\ldots,\hat{p}_k) - F(p_1,\ldots,p_k)| \leq C_B\left(h^\beta + h^{2\beta} + \frac{1}{nh^d}\right), \tag{5}$$

where $d := \max_i d_i$, and $C_V$ and $C_B$ are constant in the sample size $n$ and bandwidth $h$ for exact values. To the best of our knowledge, this is the first time an exponential inequality like (4) has been established for general density functional estimation. This variance bound does not depend on $h$ and the bias bound is minimized by $h \asymp n^{-\frac{1}{\beta+d}}$, we have the convergence rate

$$|\mathbb{E}F(\hat{p}_1,\ldots,\hat{p}_k) - F(p_1,\ldots,p_k)| \in O\left(n^{-\frac{\beta}{\beta+d}}\right).$$

It is interesting to note that, in optimizing the bandwidth for our density functional estimate, we use a smaller bandwidth than is optimal for minimizing the bias of the KDE. Intuitively, this reflects the fact that the plug-in estimator, as an integral functional, performs some additional smoothing.

We can use our exponential concentration bound to obtain a bound on the true variance of $F(\hat{p}_1,\ldots,\hat{p}_k)$. If $G : [0,\infty) \to \mathbb{R}$ denotes the cumulative distribution function of the squared deviation of $F(\hat{p}_1,\ldots,\hat{p}_k)$ from its mean, then

$$1 - G(\varepsilon) = \mathbb{P}\left((F(\hat{p}_1,\ldots,\hat{p}_k) - \mathbb{E}F(\hat{p}_1,\ldots,\hat{p}_k))^2 > \varepsilon\right) \leq 2\exp\left(-\frac{2\varepsilon n}{C_V^2}\right).$$

Thus,

$$\mathbb{V}[F(\hat{p}_1,\ldots,\hat{p}_k)] = \mathbb{E}\left[(F(\hat{p}_1,\ldots,\hat{p}_k) - \mathbb{E}F(\hat{p}_1,\ldots,\hat{p}_k))^2\right]$$
$$= \int_0^\infty 1 - G(\varepsilon)\, d\varepsilon \leq 2\int_0^\infty \exp\left(-\frac{2\varepsilon n}{C_V^2}\right) = C_V^2 n^{-1}.$$

We then have a mean squared error of

$$\mathbb{E}\left[(F(\hat{p}_1,\ldots,\hat{p}_k) - F(p_1,\ldots,p_k))^2\right] \in O\left(n^{-1} + n^{-\frac{2\beta}{\beta+d}}\right),$$

which is in $O(n^{-1})$ if $\beta \geq d$ and $O\left(n^{-\frac{2\beta}{\beta+d}}\right)$ otherwise.

It should be noted that the constants in both the bias bound and the variance bound depend exponentially on the dimension $d$. Lower bounds in terms of $d$ are unknown for estimating most density functionals of interest, and an important open problem is whether this dependence can be made asymptotically better than exponential.

## 4  Bias Bound

In this section, we precisely state and prove the bound on the bias of our density functional estimator, as introduced in Section 3.

Assume each $p_i \in \Sigma(\beta, L, r, d)$ (for $i \in [k]$), assume $f : \mathbb{R}^k \to \mathbb{R}$ is twice continuously differentiable, with first and second derivatives all bounded in magnitude by some $C_f \in \mathbb{R}$,[1] and assume the kernel $K : \mathbb{R} \to \mathbb{R}$ has bounded support $[-1, 1]$ and satisfies

$$\int_{-1}^{1} K(u)\, du = 1 \quad \text{and} \quad \int_{-1}^{1} u^j K(u)\, du = 0 \quad \text{for all } j \in \{1, \dots, \ell\}.$$

Then, there exists a constant $C_B \in \mathbb{R}$ such that

$$|\mathbb{E}F(\hat{p}_1, \dots, \hat{p}_k) - F(p_1, \dots, p_k)| \leq C_B \left( h^\beta + h^{2\beta} + \frac{1}{nh^d} \right).$$

## 4.1 Proof of Bias Bound

By Taylor's Theorem, $\forall x = (x_1, \dots, x_k) \in \mathcal{X}_1 \times \dots \times \mathcal{X}_k$, for some $\xi \in \mathbb{R}^k$ on the line segment between $\hat{p}(x) := (\hat{p}_1(x_1), \dots, \hat{p}_k(x_k))$ and $p(x) := (p_1(x_1), \dots, p_k(x_k))$, letting $H_f$ denote the Hessian of $f$

$$|\mathbb{E}f(\hat{p}(x)) - f(p(x))| = \left| \mathbb{E}(\nabla f)(p(x)) \cdot (\hat{p}(x) - p(x)) + \frac{1}{2}(\hat{p}(x) - p(x))^T H_f(\xi)(\hat{p}(x) - p(x)) \right|$$

$$\leq C_f \left( \sum_{i=1}^{k} |B_{p_i}(x_i)| + \sum_{i < j \leq k} |B_{p_i}(x_i)B_{p_j}(x_j)| + \sum_{i=1}^{k} \mathbb{E}[\hat{p}_i(x_i) - p_i(x_i)]^2 \right)$$

where we used that $\hat{p}_i$ and $\hat{p}_j$ are independent for $i \neq j$. Applying Hölder's Inequality,

$$|\mathbb{E}F(\hat{p}_1, \dots, \hat{p}_k) - F(p_1, \dots, p_k)| \leq \int_{\mathcal{X}_1 \times \dots \times \mathcal{X}_k} |\mathbb{E}f(\hat{p}(x)) - f(p(x))|\, dx$$

$$\leq C_f \left( \sum_{i=1}^{k} \int_{\mathcal{X}_i} |B_{p_i}(x_i)| + \mathbb{E}[\hat{p}_i(x_i) - p_i(x_i)]^2\, dx_i + \sum_{i < j \leq k} \int_{\mathcal{X}_i} |B_{p_i}(x_i)|\, dx_i \int_{\mathcal{X}_j} |B_{p_j}(x_j)|\, dx_j \right)$$

$$\leq C_f \left( \sum_{i=1}^{k} \sqrt{\int_{\mathcal{X}_i} B_{p_i}^2(x_i)\, dx_i} + \int_{\mathcal{X}_i} \mathbb{E}[\hat{p}_i(x_i) - p_i(x_i)]^2\, dx_i \right.$$

$$\left. + \sum_{i < j \leq k} \sqrt{\int_{\mathcal{X}_i} B_{p_i}^2(x_i)\, dx_i \int_{\mathcal{X}_j} B_{p_j}^2(x_j)\, dx_j} \right).$$

We now make use of the so-called Bias Lemma proven by [26], which bounds the integrated squared bias of the mirrored KDE $\hat{p}$ on $[0, 1]^d$ for an arbitrary $p \in \Sigma(\beta, L, r, d)$. Writing the bias of $\hat{p}$ at $x \in [0, 1]^d$ as $B_p(x) = \mathbb{E}\hat{p}(x) - p(x)$, [26] showed that there exists $C > 0$ constant in $n$ and $h$ such that

$$\int_{[0,1]^d} B_p^2(x)\, dx \leq Ch^{2\beta}. \tag{6}$$

Applying the Bias Lemma and certain standard results in kernel density estimation (see, for example, Propositions 1.1 and 1.2 of [30]) gives

$$|\mathbb{E}F(\hat{p}_1, \dots, \hat{p}_k) - F(p_1, \dots, p_k)| \leq C \left( k^2 h^\beta + k h^{2\beta} \right) + \frac{\|K\|_1^d}{nh^d} \leq C_B \left( h^\beta + h^{2\beta} + \frac{1}{nh^d} \right),$$

where $\|K\|_1$ denotes the 1-norm of the kernel. ∎

# 5 Variance Bound

In this section, we precisely state and prove the exponential concentration inequality for our density functional estimator, as introduced in Section 3. Assume that $f$ is Lipschitz continuous with constant $C_f$ in the 1-norm on $p_1(\mathcal{X}_1) \times \cdots \times p_k(\mathcal{X}_k)$ (i.e.,

$$|f(x) - f(y)| \leq C_f \sum_{k=1}^{\infty} |x_i - y_i|, \quad \forall x, y \in p_1(\mathcal{X}_1) \times \cdots \times p_k(\mathcal{X}_k)). \tag{7}$$

and assume the kernel $K \in L_1(\mathbb{R})$ (i.e., it has finite 1-norm). Then, there exists a constant $C_V \in \mathbb{R}$ such that $\forall \varepsilon > 0$,

$$\mathbb{P}\left(|F(\hat{p}_1, \ldots, \hat{p}_k) - \mathbb{E}F(\hat{p}_1, \ldots, \hat{p}_k)|\right) \leq 2 \exp\left(-\frac{2\varepsilon^2 n}{C_V^2}\right).$$

Note that, while we require no assumptions on the densities here, in certain specific applications, such as for some Rényi-$\alpha$ quantities, where $f = \log$, assumptions such as lower bounds on the density may be needed to ensure $f$ is Lipschitz on its domain.

## 5.1 Proof of Variance Bound

Consider i.i.d. samples $(x_1^1, \ldots, x_k^n) \in \mathcal{X}_1 \times \cdots \times \mathcal{X}_k$ drawn according to the product distribution $p = p_1 \times \cdots p_k$. In anticipation of using McDiarmid's Inequality [15], let $\hat{p}_j'$ denote the $j^{th}$ mirrored KDE when the sample $x_j^i$ is replaced by new sample $(x_j^i)'$. Then, applying the Lipschitz condition (7) on $f$,

$$|F(\hat{p}_1, \ldots, \hat{p}_k) - F(\hat{p}_1, \ldots, \hat{p}_j', \ldots, \hat{p}_k)| \leq C_f \int_{\mathcal{X}_j} |p_j(x) - p_j'(x)| \, dx,$$

since most terms of the sum in (7) are zero. Expanding the definition of the kernel density estimates $\hat{p}_j$ and $\hat{p}_j'$ and noting that most terms of the mirrored KDEs $\hat{p}_j$ and $\hat{p}_j'$ are identical gives

$$|F(\hat{p}_1, \ldots, \hat{p}_k) - F(\hat{p}_1, \ldots, \hat{p}_j', \ldots, \hat{p}_k)| = \frac{C_f}{n h^{d_j}} \int_{\mathcal{X}_j} \left| K_{d_j}\left(\frac{x - x_j^i}{h}\right) - K_{d_j}\left(\frac{x - (x_j^i)'}{h}\right) \right| dx$$

where $K_{d_j}$ denotes the $d_j$-dimensional mirrored product kernel based on $K$. Performing a change of variables to remove $h$ and applying the triangle inequality followed by the bound on the integral of the mirrored kernel proven in [26],

$$|F(\hat{p}_1, \ldots, \hat{p}_k) - F(\hat{p}_1, \ldots, \hat{p}_j', \ldots, \hat{p}_k)| \leq \frac{C_f}{n} \int_{h\mathcal{X}_j} \left| K_{d_j}(x - x_j^i) - K_{d_j}(x - (x_j^i)') \right| dx$$

$$\leq \frac{2C_f}{n} \int_{[-1,1]^{d_j}} |K_{d_j}(x)| \, dx \leq \frac{2C_f}{n} \|K\|_1^{d_j} = \frac{C_V}{n}, \quad (8)$$

for $C_V = 2C_f \max_j \|K\|_1^{d_j}$. Since $F(\hat{p}_1, \ldots, \hat{p}_k)$ depends on $kn$ independent variables, McDiarmid's Inequality then gives, for any $\varepsilon > 0$,

$$\mathbb{P}\left(|F(\hat{p}_1, \ldots, \hat{p}_k) - F(p_1, \ldots, p_k)| > \varepsilon\right) \leq 2 \exp\left(-\frac{2\varepsilon^2}{kn C_V^2/n^2}\right) = 2 \exp\left(-\frac{2\varepsilon^2 n}{k C_V^2}\right). \quad \blacksquare$$

# 6 Extension to Conditional Density Functionals

Our convergence result and concentration bound can be fairly easily adapted to to KDE-based plug-in estimators for many functionals of interest, including Rényi-$\alpha$ and Tsallis-$\alpha$ entropy, divergence, and MI, and $L_p$ norms and distances, which have either the same or analytically similar forms as as the functional (3). As long as the density of the variable being conditioned on is lower bounded on its domain, our results also extend to conditional density functionals of the form [2]

$$F(P) = \int_{\mathcal{Z}} P(z) f\left(\int_{\mathcal{X}_1 \times \cdots \times \mathcal{X}_k} g\left(\frac{P(x_1, z)}{P(z)}, \frac{P(x_2, z)}{P(z)}, \ldots, \frac{P(x_k, z)}{P(z)}\right) d(x_1, \ldots, x_k)\right) dz \quad (9)$$

including, for example, Rényi-$\alpha$ conditional entropy, divergence, and mutual information, where $f$ is the function $x \mapsto \frac{1}{1-\alpha} \log(x)$. The proof of this extension for general $k$ is essentially the same as for the case $k = 1$, and so, for notational simplicity, we demonstrate the latter.

## 6.1 Problem Statement, Assumptions, and Estimator

For given dimensions $d_x, d_z \geq 1$, consider random vectors $X$ and $Z$ distributed on unit cubes $\mathcal{X} := [0,1]^{d_x}$ and $\mathcal{Z} := [0,1]^{d_z}$ according to a joint density $P : \mathcal{X} \times \mathcal{Z} \to \mathbb{R}$. We use a random sample of $2n$ i.i.d. points from $P$ to estimate a conditional density functional $F(P)$, where $F$ has the form (9).

Suppose that $P$ is in the Hölder class $\Sigma(\beta, L, r, d_x + d_z)$, noting that this implies an analogous condition on each marginal of $P$, and suppose that $P$ bounded below and above, i.e., $0 < \kappa_1 := \inf_{x \in \mathcal{X}, z \in \mathcal{Z}} P(z)$ and $\infty > \kappa_2 := \inf_{x \in \mathcal{X}, z \in \mathcal{Z}} P(x,z)$. Suppose also that $f$ and $g$ are continuously differentiable, with

$$C_f := \sup_{x \in [c_g, C_g]} |f(x)| \quad \text{and} \quad C_{f'} := \sup_{x \in [c_g, C_g]} |f'(x)|, \tag{10}$$

where

$$c_g := \inf g\left(\left[0, \frac{\kappa_2}{\kappa_1}\right]\right) \quad \text{and} \quad C_g := \sup g\left(\left[0, \frac{\kappa_2}{\kappa_1}\right]\right).$$

After estimating the densities $P(z)$ and $P(x,z)$ by their mirrored KDEs, using $n$ independent data samples for each, we clip the estimates of $P(x,z)$ and $P(z)$ below by $\kappa_1$ and above by $\kappa_2$ and denote the resulting density estimates by $\hat{P}$. Our estimate $F(\hat{P})$ for $F(P)$ is simply the result of plugging $\hat{P}$ into equation (9).

## 6.2 Proof of Bounds for Conditional Density Functionals

We bound the error of $F(\hat{P})$ in terms of the error of estimating the corresponding unconditional density functional using our previous estimator, and then apply our previous results.

Suppose $P_1$ is either the true density $P$ or a plug-in estimate of $P$ computed as described above, and $P_2$ is a plug-in estimate of $P$ computed in the same manner but using a different data sample. Applying the triangle inequality twice,

$$
\begin{aligned}
|F(P_1) - F(P_2)| &\leq \int_{\mathcal{Z}} \left| P_1(z) f\left( \int_{\mathcal{X}} g\left( \frac{P_1(x,z)}{P_1(z)} \right) dx \right) - P_2(z) f\left( \int_{\mathcal{X}} g\left( \frac{P_1(x,z)}{P_1(z)} \right) dx \right) \right| \\
&\quad + \left| P_2(z) f\left( \int_{\mathcal{X}} g\left( \frac{P_1(x,z)}{P_1(z)} \right) dx \right) - P_2(z) f\left( \int_{\mathcal{X}} g\left( \frac{P_2(x,z)}{P_2(z)} \right) dx \right) \right| dz \\
&\leq \int_{\mathcal{Z}} |P_1(z) - P_2(z)| \left| f\left( \int_{\mathcal{X}} g\left( \frac{P_1(x,z)}{P_1(z)} \right) dx \right) \right| \\
&\quad + P_2(z) \left| f\left( \int_{\mathcal{X}} g\left( \frac{P_1(x,z)}{P_1(z)} \right) dx \right) - f\left( \int_{\mathcal{X}} g\left( \frac{P_2(x,z)}{P_2(z)} \right) dx \right) \right| dz
\end{aligned}
$$

Applying the Mean Value Theorem and the bounds in (10) gives

$$
\begin{aligned}
|F(P_1) - F(P_2)| &\leq \int_{\mathcal{Z}} C_f |P_1(z) - P_2(z)| + \kappa_2 C_{f'} \left| \int_{\mathcal{X}} g\left( \frac{P_1(x,z)}{P_1(z)} \right) - g\left( \frac{P_2(x,z)}{P_2(z)} \right) dx \right| dz \\
&= \int_{\mathcal{Z}} C_f |P_1(z) - P_2(z)| + \kappa_2 C_{f'} \left| G_{P_1(z)}(P_1(\cdot, z)) - G_{P_2(z)}(P_2(\cdot, z)) \right| dz,
\end{aligned}
$$

where $G_z$ is the density functional

$$G_{P(z)}(Q) = \int_{\mathcal{X}} g\left( \frac{Q(x)}{P(z)} \right) dx.$$

Note that, since the data are split to estimate $P(z)$ and $P(x,z)$, $G_{\hat{P}(z)}(\hat{P}(\cdot, z))$ depends on each data point through only one of these KDEs. In the case that $P_1$ is the true density $P$, taking the

expectation and using Fubini's Theorem gives

$$\mathbb{E}|F(P) - F(\hat{P})| \leq \int_{\mathcal{Z}} C_f \mathbb{E}|P(z) - \hat{P}(z)| + \kappa_2 C_{f'} \mathbb{E}\left|G_{P(z)}(P(\cdot,z)) - G_{\hat{P}(z)}(\hat{P}(\cdot,z))\right| dz,$$

$$\leq C_f \sqrt{\int_{\mathcal{Z}} \mathbb{E}(P(z) - \hat{P}(z))^2 dz} + 2\kappa_2 C_{f'} C_B \left(h^\beta + h^{2\beta} + \frac{1}{nh^d}\right)$$

$$\leq (2\kappa_2 C_{f'} C_B + C_f C)\left(h^\beta + h^{2\beta} + \frac{1}{nh^d}\right)$$

applying Hölder's Inequality and our bias bound (5), followed by the bias lemma (6). This extends our bias bound to conditional density functionals. For the variance bound, consider the case where $P_1$ and $P_2$ are each mirrored KDE estimates of $P$, but with one data point resampled (as in the proof of the variance bound, setting up to use McDiarmid's Inequality). By the same sequence of steps used to show (8),

$$\int_{\mathcal{Z}} |P_1(z) - P_2(z)|\, dz \leq \frac{2\|K\|_1^{d_z}}{n},$$

and

$$\int_{\mathcal{Z}} \left|G_{P(z)}(P(\cdot,z)) - G_{\hat{P}(z)}(\hat{P}(\cdot,z))\right| dz \leq \frac{C_V}{n}.$$

(by casing on whether the resampled data point was used to estimate $P(x,z)$ or $P(z)$), for an appropriate $C_V$ depending on $\sup_{x \in [\kappa_1/\kappa_2, \kappa_2/\kappa_1]} |g'(x)|$. Then, by McDiarmid's Inequality,

$$\mathbb{P}\left(|F(\hat{p}_1, \ldots, \hat{p}_k) - F(p_1, \ldots, p_k)| > \varepsilon\right) = 2\exp\left(-\frac{\varepsilon^2 n}{4C_V^2}\right). \quad \blacksquare$$

### 6.3  Application to Rényi-$\alpha$ Conditional Mutual Information

As an example, we demonstrate our concentration inequality to the Rényi-$\alpha$ Conditional Mutual Information (CMI). Consider random vectors $X, Y$, and $Z$ on $\mathcal{X} = [0,1]^{d_x}$, $\mathcal{Y} = [0,1]^{d_y}$, $\mathcal{Z} = [0,1]^{d_z}$, respectively. $\alpha \in (0,1) \cup (1, \infty)$, the Rényi-$\alpha$ CMI of $X$ and $Y$ given $Z$ is

$$I(X;Y|Z) = \frac{1}{1-\alpha} \int_{\mathcal{Z}} P(z) \log \int_{\mathcal{X} \times \mathcal{Y}} \left(\frac{P(x,y,z)}{P(z)}\right)^\alpha \left(\frac{P(x,z)P(y,z)}{P(z)^2}\right)^{1-\alpha} d(x,y)\, dz. \quad (11)$$

In this case, the estimator which plugs mirrored KDEs for $P(x,y,z)$, $P(x,z)$, $P(y,z)$, and $P(z)$ into (11) obeys the concentration inequality (4) with $C_V = \kappa^* \|K\|_1^{d_x + d_y + d_z}$, where $\kappa^*$ depends only on $\alpha$, $\kappa_1$, and $\kappa_2$.

## Footnotes

[1]If $p_1(\mathcal{X}_1) \times \dots \times p_k(\mathcal{X}_k)$ is known to lie within some cube $[\kappa_1, \kappa_2]^k$, then it suffices for $f$ to be twice continuously differentiable on $[\kappa_1, \kappa_2]^k$ (and the boundedness condition follows immediately). This will be important for our application to Rényi-$\alpha$ Conditional Mutual Information.

[2]We abuse notation slightly and also use $P$ to denote all of its marginal densities.

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
