[Reviews · NeurIPS 2014]

Submitted by Assigned_Reviewer_15

Review for "Exponential Concentration of a Density Functional Estimator"

This paper derives an exponential concentration inequality for a plug-in estimator of a class of integral functionals of one or more continuous probability densities, which includes entropy, divergence, mutual information, and others. From the concentration inequality and an analysis of the bias, mean squared error convergence rates of the estimator are derived. It is then shown how the concentration inequality can be used to find bounds on the error of an estimator for conditional mutual information.

This work could be significant in that the results can be applied to a large class of integral functionals of probability densities. Some of the methods used here may also provide inspiration for generalizing other existing entropy or divergence estimators to this class.

However, the mathematical derivations in this paper are not polished and cause confusion. The case where this is most harmful is in the derivation of Eq (8). The kernel K_{d_j} is never defined, nor is it obvious from looking at [25]. Even if it denotes the product kernel, it is not clear how the second line follows from the first line in this set of equations. Thus it is unclear how the bound on the integral of the mirrored kernel is applied. Since Eq. (8) is crucial to the concentration inequality, it is important for this to be clarified. Other minor mistakes and notational issues are spread throughout the paper, some of which are described below.

Minor comments
Lines 69-71: Repeated phrase "we can." Also the last sentence in this paragraph does not make sense.

Section 1.2: Another recent k-nn based estimator of divergence that converges at the parametric rate for \beta > d is worth mentioning:

K. Moon and A. O. Hero III, "Ensemble estimation of mutlivariate f-divergence," In International Symposium on Information Theory, 2014.

Another important work that should be mentioned is:

X. Nguyen, M. J. Wainwright, and M. I. Jordan, “Estimating divergence functionals and the likelihood ratio by convex risk minimization,” IEEE Trans. Inform. Theory, vol. 56, no. 11, pp. 5847–5861, 2010.

These estimators are among the few estimators of divergence where convergence rates are known.

Line 124: Change \alpha_d to i_d.

Section 4.1: There are some notational inconsistencies and errors in this section. For example, \tilde{p_h} is first defined as a mapping to the real line, but then in line 242 it seems to map to a k dimensional space. There are also multiple cases where \tilde{p_h} should be replaced with \tilde{p_i}. Also lots of x's should be replaced with x_i for notational consistency with other sections. Although it was possible for me to figure out the proof here, the errors made it more difficult.

Equations (9), (10), and (11): I believe there should be a 1/(1-\alpha) term in each of these equations.

Line 346: The last sentence in this paragraph references Eq. (10) but should reference Eq. (9) instead.

Lines 368, 374, and 377: The \hat{I}'s and f's should have \alpha in the subscript.

Lines 368-377: Another step in this derivation would help clarify it, especially since there is room for it in the paper.

Lines 446-447: The reference [26] should be updated as it has been published: "Sricharan, Kumar, Dennis Wei, and Alfred O. Hero. "Ensemble estimators for multivariate entropy estimation." Information Theory, IEEE Transactions on 59.7 (2013): 4374-4388."
Summary: The results of this paper, if correct, are a valuable contribution to the problem of estimating an integral functional of one or more densities. While the theory presented in the paper is plausible, I was unable to verify the proof of the main result due to missing information, typos, and notational inconsistencies.

The author response to the reviews, however, gave me additional confidence in the correcteness of the results.

Submitted by Assigned_Reviewer_31

QUALITY

The results are interesting, relevant for practice (e.g. for designing conditional mutual independence tests) and the technical development is o.k. as far as I could check. I do have two issues though (one which is related to the 'as far as I could check' - there are various unclarities in the math part of the paper, see below).

(1) you claim that your results can be applied to Renyi and hence KL divergences, but these are *not* of the form (3) because p_1 and p_2 are defined on the same set X_1 there , so taking a product integral makes no sense. I suspect that this can be handled using similar techniques as you use in Section 6 for CMI estimation, which is also of different form; but you should definitely point out in the beginning of the paper that some modifications are needed; this had me confused for quite a while.

(2) there is a step in the argument in Section 6 (which is crucial given the paper's motivation) which I really can't follow, on the bottom of page 7: first, where does the |1 - \alpha| in the formula come from/go to? Second, it seems you use the triangle inequality and the mean value theorem in a single step. This needs to be separated (and probably decomposed in smaller steps even further). Where does the \kappa_2 and \log \kappa* come from? I really don't get it. In fact my recommendation below is conditional on getting more information about this derivation in the author feedback.

CLARITY

The goals and presentation are clear. I do have an issue with the notation of the main formula, i.e. the display on page 1 and eq. (3): why do you write p_1(x), ..., p_k(x) there and not p_1(x_1), ..., p_k(x_k) and then explain that the x in dx stands for x = (x1, ..., x_k). The current definition makes, strictly speaking, no sense (p_i is a function of x_i, not x !). Am I correct or did I miss something? (Again my positive recommendation below is conditional on getting a reply to this question!)

ORIGINALITY

Moderate - this is a step further in a direction of which it has recently become apparent that it is both significant and that interesting results are obtainable.

SIGNIFICANCE

Substantial - it is definitely useful to have estimates with exponential concentration.

FURTHER REMARKS

- Why is eq. (11) needed? Isn't this just a repetition of (9)? Better use this space for more explanation about what goes on further below...

- When you refer to Renyi divergences: perhaps good to quote Van Erven and Harremoes' recent IEEE paper, which contains 'all you ever wanted to know about Renyi divergence' and seems to be rapidly becoming a standard reference (note I am neither Van Erven nor Harremoes)

- I wonder whether the following papers are relevant, you might want to check them out. They deal with countably infinite outcome spaces but otherwise similar concerns as yours, and predate most of what you cite (again, I am not one of the authors :-).

On the lower limits of entropy estimation.
AJ Wyner, D Foster
submitted to IEEE Transactions on Information Theory (apparently never officially published!?)

Nonparametric entropy estimation for stationary processes and random fields, with applications to English text
I Kontoyiannis, PH Algoet, YM Suhov, AJ Wyner
Information Theory, IEEE Transactions on 44 (3), 1319-1327

- The final sentence of the second paragraph of page 2 is grammatically incorrect and incomprehensible

UPDATE

After having seen the author's response, I updated my score to a 7. I would still like to insist though that my issue with Eq. (3) (see under CLARITY above) is explicitly dealt with in the final paper; either correct the formula (if I am correct) or explain in more detail how to read the formula (if I was wrong and the formula was correct) (the authors remained silent about this in their reply).
Summary: Interesting, significant and presumably technically correct paper, but some sloppiness in formulation of main definition and one of the proofs.

Submitted by Assigned_Reviewer_37

This paper analyzes a plug-in estimator for a large class of integral functionals of one or more continuous probability densities. This class includes important families of entropy, divergence, mutual information, and their conditional versions. For densities on the d-dimensional unit cube [0,1]^d that lie in a \beta-Hoelder smoothness class, this paper proves the estimator
converges at the rate O(n^{-d/(d+\beta)}), and that it obeys an exponential concentration inequality about its mean.

Quality: Fine.

Clarity: Although this paper deals with a complicated matrial, the manuscript is well-organized and easy to read.

Originality:
The result is almost similar to [25]. In fact,
1. both apply the same kernel (mirrored KDE on [0,1]^d), assuming \beta-Hoelder smoothness class.
2. both derive the results in Sections 4 and 5 in an essentially similar way akthough the authors claim that the current paper simplifies and generalizes the previous one. In my understanding, the differences is that the current paper seeks generalization for all the forms expressed by (3). The derivation seems to be straightforward.

Significance:
This paper assumes \beta-Hoelder smoothness, so the derivation is rather standard. However, this paper is informative and pleasing theoretician
Summary: This paper may be accepted unless there are many other good papers.
Author Feedback
Author rebuttal: We thank the reviewers for their valuable suggestions and comments.

*** Three major issues pointed out by reviewers: ***

** 1. Regarding the derivation of the Variance Bound (Eq. 8): **

The kernel $K_{d_j}$ indeed denotes the $d_j$ dimensional product kernel. [25] showed that the integral of the mirrored kernel over the unit cube X_j is precisely the 1-norm of the kernel, because the reflected components precisely account for the mass of the kernel that lies outside X_j. Hence, we can replace the integral over X_j of the sum (over reflections) of kernels with the integral of the original (product) kernel over R^{d_j}. There is a mistake in the second and third lines that might be the cause of confusion - the integral should be over R^{d_j} rather than over X_j. We fixed this and added an intermediate step after the first line which includes the sum of the reflected kernels (this step was omitted initially, because it required defining notation used in [25] that we did not wanted to introduce explicitly, but the reviewers fairly point out that this is confusing).

** 2. Regarding the application to Renyi CMI: **

a) The 1/|1 - alpha| is missing from the definition of Renyi CMI (Eqs. 9, 10, 11).

b) The combination of the triangle inequality and the mean value theorem understandably also caused confusion. The triangle inequality is used here in the following form:

|ab - a'b'| = |ab - ab' + ab' - a'b'|
<= |ab - ab'| + |ab' - a'b'|
= |a||b - b'| + |a - a'||b'|

Here, a = P(z), a' = P'(z), and b and b' are the log(...) terms. Hence, we bound |a| <= kappa_2 and |b'| <= |log(kappa_*)|. We then apply the mean value theorem to the |b - b'| and |a - a'| terms (the former results in the kappa^* constant).

We initially refrained from including the intermediate step, because using the notation in the paper each step above took 4 complete lines to write out. However, we agree that too many steps are being skipped here, and have inserted some notation similar to the above to express these steps more compactly and clearly.

** 3. Regarding the Renyi and KL divergence as Density Functionals **

The reviewer correctly points out that divergences require a joint integral, not a product integral, of the two distributions, and hence are not of the form (3). This has been clarified in the beginning of the paper and in Section 6.

*** Two points regarding originality: ***

1. While we follow the same framework as suggested by [25], our paper is about adapting these steps to a broad class of density functionals characterized by (a corrected version of) equation (3), and showing, how this in turn leads to interesting results for complex quantities such as Renyi conditional mutual information.

To the best of our knowledge, our submission is the first paper that proves finite sample exponential concentration bound for conditional mutual information and conditional divergences.

Such quantities are relevant to many in the NIPS community. However, there has been relatively little theoretical work on measuring or testing for conditional(in)dependence in a continuous setting, due largely to the sheer mathematical complexity of the quantities involved.

2. While the beta-Holder assumption is indeed a standard one for bounding the bias of KDEs via a fairly standard derivation, our paper simply cites this part of the derivation (the ``Bias Lemma'') from [25] and focuses instead on how this translates into a bound on the bias of the plug-in estimate, and on deriving the variance bounds in the general case and the Renyi CMI case. Furthermore, the Holder assumption is relevant only to the bias bound, and not to the variance bound or the application to Renyi CMI estimation. Indeed, one unmentioned development in our paper (compared to [25]) is that we segregate the assumptions for the Bias Bound (including the Holder and Boundary conditions) from those needed for the Variance Bound. Hence, we suggest a way of proving an exponential concentration inequality not only for our mirrored-KDE-based plug-in estimator, but for a wide range of plug-in estimators based on variants of KDEs.

*** Other minor corrections/clarifications ***

1. The end of paragraph 2 on page 2 (lines 69-71) had errors. It now reads:

For example, [14] shows how this can be applied to optimally analyze forest density estimation algorithms. We can also bound the distribution of the CMI estimator when the true CMI is 0. Consequently, we can use this estimator to devise a hypothesis test for conditional independence with bounded type I error probability.

2. Thanks to the reviewers for pointing out several minor errors in the paper, and for suggesting some additional related work. We have corrected these errors and mentioned these references in the Related Work section.